# Stratification of Colorectal Patients Based on Survival Analysis Shows the Value of Consensus Molecular Subtypes and Reveals the CBLL1 Gene as a Biomarker of CMS2 Tumours

**DOI:** 10.3390/ijms25031919

**Published:** 2024-02-05

**Authors:** Gloria Alfonsín, Alberto Berral-González, Andrea Rodríguez-Alonso, Macarena Quiroga, Javier De Las Rivas, Angélica Figueroa

**Affiliations:** 1Epithelial Plasticity and Metastasis Group, Instituto de Investigación Biomédica de A Coruña (INIBIC), Complexo Hospitalario Universitario de A Coruña (CHUAC), Sergas, Universidade da Coruña (UDC), 15006 A Coruña, Spain; maria.gloria.alfonsin.rey@sergas.es (G.A.); andrea.rodriguez.alonso@sergas.es (A.R.-A.); macarena.quiroga.fernandez@sergas.es (M.Q.); 2Bioinformatics and Functional Genomics Group, Cancer Research Center (CiC-IBMCC, CSIC/USAL & IBSAL), Consejo Superior de Investigaciones Cientificas (CSIC), University of Salamanca (USAL) and Instituto de Investigación Biomédica de Salamanca (IBSAL), 37007 Salamanca, Spain; aberralgonzalez@usal.es

**Keywords:** cancer stem cells, CBL proteins, colon cancer, consensus molecular subtypes, epithelial-to-mesenchymal transition, Hakai, Wnt

## Abstract

The consensus molecular subtypes (CMSs) classification of colorectal cancer (CRC) is a system for patient stratification that can be potentially applied to therapeutic decisions. Hakai (CBLL1) is an E3 ubiquitin–ligase that induces the ubiquitination and degradation of E-cadherin, inducing epithelial-to-mesenchymal transition (EMT), tumour progression and metastasis. Using bioinformatic methods, we have analysed CBLL1 expression on a large integrated cohort of primary tumour samples from CRC patients. The cohort included survival data and was divided into consensus molecular subtypes. Colon cancer tumourspheres were used to analyse the expression of stem cancer cells markers via RT-PCR and Western blotting. We show that CBLL1 gene expression is specifically associated with canonical subtype CMS2. WNT target genes LGR5 and c-MYC show a similar association with CMS2 as CBLL1. These mRNA levels are highly upregulated in cancer tumourspheres, while CBLL1 silencing shows a clear reduction in tumoursphere size and in stem cell biomarkers. Importantly, CMS2 patients with high CBLL1 expression displayed worse overall survival (OS), which is similar to that associated with CMS4 tumours. Our findings reveal CBLL1 as a specific biomarker for CMS2 and the potential of using CMS2 with high CBLL1 expression to stratify patients with poor OS.

## 1. Introduction

Colorectal cancer (CRC) is the third most frequent type of cancer and the second major cause of death from cancer, which is mainly due to metastasis. Indeed, around 20–30% of the patients are metastatic at their initial diagnosis and around 50% of the patients will develop metastasis during the course of the disease [1]. CRC is a very heterogeneous malignancy, and this variability is considered a major clinical challenge. At present, only a few genetic alterations, such as KRAS, BRAF mutations, MSI and CIMP status, lead to specific decisions for therapeutic interventions. Recognising the significance of molecular traits in predicting prognosis and drug response and acknowledging the inadequacy of current classifications to address clinical needs [2], it has become imperative to propose a molecular classification for CRC. This study aims to streamline the translation of molecular characteristics into practical applications in the clinic. Guinney et al. [3] described the existence of four consensus molecular subtypes (CMSs) for CRC, which, at present, represents the most robust system to obtain the stratification of CRC patients and allow further advances in therapeutic decisions based on the activity (expression) of certain gene sets (assigned to each CMS) combined with different molecular and clinical features [4]. Four CMSs have been identified in CRC: CMS1, 2, 3 and 4. CMS1 or MSI immune (15% of the early-stage CRC tumours) is characterised by hypermutation, with an enrichment for BRAFV600E mutations, hypermethylation, microsatellite instability and a strong infiltration with activated immune cells. CMS2 or the canonical subtype (40% of early-stage tumours) are epithelial tumours with the activation of WNT and MYC as well as EGFR signalling and a high expression of cyclins. CMS3 or the metabolic subtype (13% of early-stage tumours) has an evident metabolic dysregulation, including the activation of glutaminolysis and lipogenesis, and a fewer copy number alterations. Finally, CMS4 or the mesenchymal subtype (25% of early-stage tumours) is characterised by the activation of pathways related to epithelial–mesenchymal transition (EMT) and stemness, and a high overexpression of extracellular proteins related to TGF-β activation, integrin, stromal invasion, and angiogenesis [3,4].

At early stages of tumour progression in CRC, a process named epithelial-to-mesenchymal transition (EMT) occurs, which is characterised by the loss of cell–cell contacts between epithelial cells, the loss of cell adhesion to the substrate, and the disruption of cell polarity. The activation of the EMT process is associated with the acquisition of invasive and migratory capabilities and, as a consequence, cells become metastatic. The most established biomarker of the EMT is the loss of E-cadherin at cell–cell contacts, which is accompanied by the gain of mesenchymal markers including Vimentin and N-cadherin among others [5,6]. EMT is an important process in the acquisition and maintenance of stem cell-like properties. Indeed, cancer stem cells often exhibit EMT characteristics, which has implications in metastasis and therapy resistance [7]. Therefore, the EMT has emerged as an attractive target for cancer therapy [8,9]. In fact, a significant amount of evidence shows that conventional therapies generally fail in cancer cells that have entered the cancer stem-cell state through the activation of the EMT process [10,11]. On the other hand, the alteration in the Wnt/β-catenin signalling pathway in colorectal cancer stem cells (CSCs) is well described [12]. The best-established CSC biomarker for colorectal is probably Lgr5, which is required for the maintenance of spheroid-derived colon cancer stem cells [13,14,15,16]. In fact, it has been proposed that the EMT is mediated by Lgr5 via Wnt/β-catenin signalling [17]. Although Lgr5 is a specific biomarker for colorectal CSCs, there are other universal CSC markers including the transcription factors Sox2, c-Myc, Klf4, Oct4, and Nanog [18].

During the EMT process, the E3 ubiquitin–ligase Hakai regulates E-cadherin expression at a post-translational level. The Hakai protein, encoded by the human CBLL1 gene, induces the ubiquitination and degradation of E-cadherin and, as a consequence, its disappearance at cell–cell contacts, impacting tumour progression and metastasis. Hakai interacts and ubiquitinates E-cadherin when E-cadherin is tyrosine phosphorylated by Src [19,20]. Hakai is highly expressed in human colon and gastric adenocarcinomas compared to normal healthy adjacent tissues [20,21]. Many E3 ubiquitin ligases have been reported to play a crucial role in the EMT and in CSCs characteristics [22,23]. However, the role of Hakai in CSC is still unknown. There are different families of E3 ubiquitin–ligases including the HECT-domain family, the RING-finger domain, and the RING-between-RING ligases (RBR), which share features of the RING and HECT families [24]. The most abundant family is the RING-type domain E3 ubiquitin ligases that need phosphorylated target substrates for recognition. Cbl is a RING-type E3 ubiquitin–ligases subfamily that depends on phosphorylation at a tyrosine residue (pTyr). There are three members of this family: Cbl, Cbl-b and Cbl-c. Hakai, also named Cbl-like-1 or Cbll1, was firstly included as a member of the Cbl family. However, the posterior elucidation of its molecular structure showed that Hakai is not a typical Cbl protein. Hakai contains a unique and novel HYB (Hakai pTyr-binding) domain that forms a phosphotyrosine-binding pocket, consisting of a pair of monomers arranged in an anti-parallel configuration, which converts Hakai in an excellent drug target against cancer [25,26].

Given the potential clinical application of the consensus molecular classification system for predicting prognosis and response to therapy in CRC patients, the aim of this study was to investigate the potential clinical validity of the CBLL1 gene (Hakai protein) to predict prognosis based on CMS classification. In this way, we show that the high expression of the CBLL1 gene is specifically associated with the CMS2 canonical subtype, whereas in comparison, the rest of the members of the CBL protein family do not show such an association. Similarly, the same association as CBLL1 was found for the LGR5 gene, a cancer stem cell biomarker in colon cancer, as well as with the transcription factor c-MYC, which is a well-known universal marker of CSCs. We also show that CBLL1, LGR5, and c-MYC mRNA levels are highly upregulated in cancer stem cell tumourspheres compared to monolayer cell cultures, while silencing CBLL1 in cancer stem cell tumourspheres results in a significant reduction in the size of the tumoursphere, which is accompanied by the downregulation of stem cell biomarkers. Importantly, we show that high CBLL1 expression was significantly associated with shorter overall survival (OS) and poor prognosis in the CMS2 subpopulation. The same behaviour is observed for the CSC biomarker LGR5 in the CMS2 subset of CRC patients. Furthermore, CMS2 patients with high CBLL1 expression and poor survival showed the worse OS, as did tumour samples that were classified as CMS4. Considering these results together, we highlight the potential use of CBLL1 as a biomarker of CMS2 colorectal cancer patients and its implication of in the cancer stem cell tumoursphere. In addition, we show that CMS2 tumours with higher CBLL1 expression may be used for the stratification of cancer patients with a poor overall survival (OS).

## 2. Results

### 2.1. High CBLL1 Gene Expression Is Specifically Associated with the Consensus Molecular Subtype 2 (CMS2) in Colorectal Cancer Patients

The human CBLL1 gene encodes the Hakai protein. It has been reported that the Hakai protein is highly expressed in colon and gastric cancer tissues compared to normal epithelial tissues [20,27]. Furthermore, Hakai expression progressively increases in adenoma (non-malignant tissue) and in different TNM stages (I–IV) of colon cancer patients compared to adjacent colon healthy tissues, highlighting the potential of Hakai as a biomarker for CRC progression [22]. However, in recent years, it has become apparent that the consensus molecular classification of CRC subtypes, based on gene expression and clinical features, has been shown to be a well-accepted source for CRC classification [3]. To determine whether CBLL1 gene expression could be associated with one of the four consensus molecular CRC subtypes, we used robust bioinformatics and machine learning methods. Firstly, we used a large integrated cohort of 1273 primary tumour samples that include normalised transcriptomics and survival data, as previously reported [28]. As shown in the heatmap in Figure 1a, we selected from this cohort a subset including 849 tumours that were correctly assigned to one of the four consensus molecular subtypes of CRC (CMS1, 2, 3, 4). CRC tumour samples were classified using the algorithms described in the Methods section, applying a cut-off of *p*-value ≤ 0.05 (as indicated at the bottom of Figure 1a) [29]. As shown, different specific gene sets are used to classify each sample in a CMS subtype: 127 genes characterised the expression profile for CMS1, 82 genes characterised the expression profile for CMS2, 64 genes characterised the expression profile for CMS3, and 210 genes characterised the expression profile for CMS4. Once the samples were classified into CMS1 to CMS4, we investigated whether the expression of CBLL1 was associated with any of the four CMSs. Interestingly, high levels of CBLL1 expression were found to be specifically associated with the canonical subtype CMS2 (Figure 1b), which is characterised by an epithelial phenotype, upregulation of the WNT and MYC signalling pathways and chromosomal instability. However, as shown in Appendix A, no significant differences were detected when comparing the mRNA expression of CBLL1 in different TNM stages from samples of CRC patients. In addition, we further analysed the association between CBLL1 expression and the site of origin of the CRC tumours. It is well established that depending on the location of the tumours (right or left side of the colon), different molecular characteristics are detected as well as different behaviour in disease progression, overall survival and response to therapy [30]. Interestingly, CBLL1 is significantly upregulated in left-sided tumours, which have a better prognosis in late stages (III and IV) and tend to have liver and lung metastases, than in right-sided tumours, which tend to have a worse prognosis with frequent peritoneal carcinomatosis (Figure 1c) [31,32]. Consistently, higher CBLL1 expression is associated with lower risk CRC (Figure 1d). Finally, we found no significant differences when comparing CBLL1 expression by gender (Figure 1e).

We then investigated whether we could find a similar specific association between high expression of the CBLL1 gene and CMS2 in the other reported mammalian homologues of the CBL family, including CBLL2, CBL, CBLB and CBLC. As shown in Figure 2, the high expression of the rest of the protein members belonging to the CBL family was not specifically associated with CMS2. For example, the expression CBLL2, the closest family member to CBLL1 sharing a similar molecular structure, is not specifically associated with any of the four consensus molecular CRC subtypes (Figure 2a). On the contrary, high CBL expression is specifically associated with CMS4 compared to the rest of the CMSs, whereas the low expression of CBL was associated with CMS3 (Figure 2b). Interestingly, low CBLB expression was specifically associated with CMS2 (Figure 2c). Finally, the highest expression of CBLC was associated with CMS3, while low expression was associated with CMS4 (Figure 2d). Taken together, these results indicate that a high expression of CBLL1 and low expression of CBLB are associated with CMS2; a high expression of CBLC and low expression of CBL are associated with CMS3; and a high expression of CBL is associated with CMS4. In conclusion, we propose CBLL1 as a new biomarker gene for the stratification of CMS2 colorectal cancer patients.

### 2.2. Impact of Differential Expression of Specific Genes Assigned to CMS2 on Cancer Stem Cells

It is well documented that CMS2 colorectal tumours are associated with activation of the Wnt-β-catenin and Myc pathways [4]. Indeed, the Wnt pathway is thought to be responsible for the initiation of CRC in many cases, as it is involved in promoting cancer stem cell renewal and cell proliferation [33]. The classification of the CRC samples into CMS subtypes presented in the previous section (performed as described in the Methods section) was used to perform the comparison of the CMS2 tumours versus the rest of CMS tumours (1, 3, and 4). According to this classification, we had 304 CRC tumour samples assigned to CMS2 (out of a total of 849 samples; 35.8%) and 545 samples assigned to the rest of the CMS (1, 3, and 4, 64.2%). As shown in Figure 3a, the newly identified biomarker CBLL1 was significantly upregulated in CMS2 samples compared to the rest of the consensus CRC molecular subtypes. This regulation was also found in representative upregulated genes for CMS2, including the WNT pathway target gene LGR5 (Figure 3b), which is an important marker of colon cancer stem cells, and the oncogenic transcription factor MYC, which regulates tumour growth, differentiation, and proliferation (Figure 3c). To gain a comprehensive view of the potential effect of the differential gene expression of CBLL1 assigned to CMS2, we focused our attention on the potential new role of CBLL1 in regulating cancer stem cell properties.

To understand the potential molecular mechanism by which CBLL1 might be involved in cancer stem cells, HCT116 colon cell lines were grown in 2D as a monolayer cell culture (Figure 4a, left panel) and as a 3D cancer stem cell tumoursphere-derived HCT116 (Figure 4a, right panel) by culturing in ultra-low attachment plates grown in a specific medium to enrich cancer stem cell growth. As shown in Figure 4b, CBLL1 mRNA levels are highly upregulated in cancer stem cell tumoursphere-derived HCT116 compared to HCT116 monolayer cell culture. Furthermore, the high CBLL1 expression observed in the tumourspheres was accompanied by the upregulation of other established cancer stem cell markers, including LGR5, NANOG, SOX2, KLF4 and c-MYC mRNA levels (Figure 4c). This increase in cancer stem cell biomarkers was confirmed at the protein level, as observed in cancer stem cell tumourspheres for Hakai (CBLL1), Lgr5 and Nanog compared to protein expression in monolayer cultures (Figure 4d). Taken together, these results suggest a possible role for CBLL1 in the formation and/or maintenance of cancer stem cell tumourspheres.

We then tested the effect of CBLL1 silencing on HT29 cancer stem cell tumourspheres using a previously reported inducible system of viral transduction of shRNA-CBLL1 silencing [28]. CBLL1 silencing strongly affects the phenotype of the tumourspheres, including a significant reduction in tumoursphere size (Figure 5a). CBLL1 silencing was confirmed at the mRNA level (Figure 5b), and mRNA downregulation of other known stem cell markers including LGR5, NANOG and c-MYC was also detected in sh-CBLL1 HT29 cancer stem cell tumourspheres compared to sh-control tumourspheres (Figure 5c–e). Furthermore, the downregulation of Hakai (CBLL1 gene) was also confirmed at the protein level, which was accompanied by the downregulation of other stem cell markers, including Lgr5, Nanog and Klf4 (Figure 5f). These results suggest that Hakai is directly involved in the formation of cancer stem cell tumourspheres.

### 2.3. Analysis of Survival of CRC Samples Classified in CMS2

Guinney et al. previously found different associations between CMS groups and clinical variables, including overall survival (OS) [3]. Although subsequent publications show controversial data regarding the OS between different CMS, in a recent study, we highlighted the differences in prognosis observed between CMSs (taking into account gene expression signatures), confirming the clinical relevance of the biological processes implicated in each CMS [34,35]. In light of this, we aimed to investigate the potential impact of CBLL1 as a prognostic marker (measuring OS). We collected genome-wide expression data and clinical survival data for this large CRC cohort to enable accurate survival analysis and patient risk prediction. We first analysed the expression levels of CBLL1 considering the whole cohort of 849 CRC samples (Figure 6a, left panel). The samples were divided into two groups. In one group, the samples were identified as having high CBLL1 expression levels (in red), and in the other group, the samples were identified as having low expression of the CBLL1 gene (in green). We plotted two Kaplan–Meier curves over these groups of patients (Figure 6a, right panel) to see if there was a significant difference in their survival. To perform these analyses in a robust way, we use a bioinformatic algorithm described in [36] that optimises the separation of tumour samples into two groups based on the expression level (high expression versus low expression) of a query gene, using bootstrapping to identify the best expression cut-off to divide the samples, and then using a univariate Cox proportional hazards regression algorithm to calculate the hazard ratio (HR) and the significance of the difference between the two survival curves. All this was completed as described in [36]. The results show that CBLL1 overexpression was significantly associated with longer OS. However, when we analysed the expression levels of CBLL1 considering only CMS2 CRC patients (Figure 6b), and these patients were grouped into high and low CBLL1 expression (Figure 6b, left panel), the Kaplan–Meier analysis of CMS2 showed the opposite result (Figure 6b, right panel). This surprising result indicates that high CBLL1 expression was significantly associated with shorter OS and therefore associated with poor prognosis in CMS2 tumours. The same behaviour was observed when comparing LGR5 expression levels in the CRC cohort considering all patients or only CMS2 patients (see plots in Figure 6c,d). These results further suggest that CBLL1 and LGR5 can be considered as specific biomarkers of poor prognosis in CRC patients with subtype 2 (CMS2). Furthermore, this molecular stratification of CRC patients may help to identify patients with a higher likelihood of developing tumour relapse or recurrence, which will be patients with CMS2 class tumours. In addition, these biomarkers may be used to predict response to therapy and to develop novel therapeutic strategies directed to CRC CMS2 patients.

It has been reported that CMS4 tumours displayed the worst OS of all CMS [3]. We decided to investigate whether high CBLL1 expression in CMS2 patients could be a biomarker for worse prognosis and whether it was comparable to the survival observed in CMS4 tumours. As shown, no very significant differences were found between OS observed in CMS4 patients and CMS2 patients with a high CBLL1 expression (Figure 7a). However, a significant difference in OS was observed when comparing CMS4 patients with CMS2 patients with low CBLL1 expression, which showed a significantly better survival (Figure 7b). These results further suggest that the CBLL1 expression level in CMS2 is a biomarker of worse prognosis, as it can differentiate between patients with worse OS (CBLL1 high expression) and better OS (CBLL1 low expression). Furthermore, when we considered CMS2 patients that present poor survival but also high CBLL1 expression, we did not find any significant difference in the survival (OS) compared to the survival (OS) of CMS4 patients (Figure 7c); whereas CMS2 patients with good survival and low CBLL1 expression showed a much better survival (OS) than CMS4 patients (Figure 7d). In conclusion, our results underline that CMS2 patients with high CBLL1 expression and poor survival showed the worst prognosis (like CMS4 tumours), further highlighting CBLL1 expression as a clear marker of CMS2 patients with worse prognosis.

We finally wanted to investigate the mutation profile of the CBLL1 gene. By using the COSMIC database (https://cancer.sanger.ac.uk/cosmic, the Catalogue Of Somatic Mutations In Cancer, accessed on 10 February 2023), which collects somatic mutations from The Cancer Genome Atlas (TCGA) as well as from many smaller scale studies, we analysed 40,568 tumour samples, and we have not found a significant mutation rate for the CBLL1 gene in all types of tumours analysed. Indeed, 0.986% mutations were found in CBLL1, and only 0.44% corresponded to CBLL1 mis-sense mutations in all types of tumours (Appendix A). Moreover, when we analysed the presence of mis-sense mutations in 2844 CRC samples, only 1.05% corresponded to this kind of mutations (Appendix A). Then, we analysed the mutations presented in different CBLL1 domains, and we did not find a significant mutation rate in any of the domains. No significant mutations were detected in the domains involved in the E3 ubiquitin–ligase activity (0.02% mutation in pTyrB-domain and 0.005% mutations in the RING-finger domain; see Appendix A). These results highlight the absence of significant mutations in CBLL1 gene in any type of tumour. Finally, we analyse the association between CBLL1 expression and frequent gene mutation in CRC, including KRAS and BRAF. Interestingly, using a cohort of 205 samples of CRC patients, including 85 tumour samples with KRAS mutations and 120 tumour samples with KRAS wild type (non-mutated), we detected an association of CBLL1 gene expression with non-mutant KRAS samples, as shown in the boxplot (Appendix A). In a similar manner, CBLL1 gene expression was associated with non-mutant BRAF samples when analysing 205 samples of CRC patients (20 tumour samples with BRAF mutations and 195 tumour samples with non-mutated BRAF). Taken together, our results underscore the potential use of CBLL1 gene expression as a novel biomarker for CMS2 tumour samples in colon cancer patients. Our results will lead to a better patient stratification, and they can also help with identifying patients at risk and personalising therapy in CRC tumours.

## 3. Discussion

In this study, by using robust bioinformatics and machine learning methods, we provide evidence that high CBLL1 mRNA expression is specifically associated with consensus molecular subtype 2 (CMS2) in colorectal cancer patients, which is characterised by strong activation of the WNT and MYC pathways. This specific association between high CBLL1 expression and CMS2 is not found when comparing the other reported mammalian homologues of the CBL family, including CBLL2, CBL, CBLB and CBLC (Figure 1 and Figure 2). Furthermore, when we compare the CMS2 tumours with the rest of the CMS tumours, we find the same association with specific biomarkers of WNT and MYC signalling. In fact, a high expression of CBLL1 and WNT target genes, including LGR5 and c-MYC mRNA, are also specifically associated with CMS2 (Figure 3). In support of these results, by using ultra-low attachment plates and a specific medium to enrich cancer stem cells, we grow 3D tumourspheres of colon cancer stem cells and find high mRNA levels of CBLL1, LGR5, and c-MYC compared to those levels in 2D monolayer cell cultures. Moreover, CBLL1 silencing on cancer stem cell tumourspheres induces a significant reduction in tumoursphere size, which is accompanied by the downregulation of the stem cell biomarkers LGR5 and c-MYC at both mRNA and protein levels. To our knowledge, this is the first reported evidence showing that the Hakai protein is associated with colon cancer stem cells. Importantly, by grouping CMS2 patients according to high and low CBLL1 expression, the analysis of the OS shows that high CBLL1 expression is associated with shorter overall survival and poor prognosis. We show a similar behaviour for the CSC biomarker LGR5 in the same cohort of CMS2 colon cancer patients. However, we find an opposite result when analysing CBLL1 mRNA expression in the whole cohort of CRC patients (including all CMSs). We postulate that CBLL1 is a new biomarker gene for the stratification of CMS2 colorectal cancer patients and has a well-defined prognostic value in this cohort of patients.

In recent years, a deep understanding of the molecular characteristics of CRC has allowed clinicians to make better therapeutic decisions based on biomarker selection. The recently proposed CMSs for CRC is based on the association between biological characteristics, including mRNA expression and molecular and clinical features [3]. At present, this CMS classification represents the most important attempt to clinically understand the heterogeneity of CRC. The four CMSs include CMS1, which represents the immune phenotype and shows a hypermutated MSI; CMS2, which is the canonical subtype, with an upregulation of WNT and MYC signalling; CMS3 or the metabolic subtype, which provokes deregulation of metabolic genes; and CMS4 or the mesenchymal subtype, which is involved in a high expression of EMT genes, activation of TGF-β signalling and induction of angiogenesis and metastasis [3,4]. Considering also the functional context in cancer, Hakai (CBLL1 gene) has been implicated in tumour progression and metastasis through a possible role in the EMT programme. Hakai binds to phosphorylated E-cadherin by Src and induces its degradation, thereby inducing the EMT programme by disrupting cell–cell contacts. Given the well-established molecular mechanism by which Hakai is involved in EMT, we expected to find the CBLL1 gene (Hakai protein) as a biomarker of CMS4, which represents the mesenchymal subtype and has a high expression of EMT genes. Unexpectedly, we identified CBLL1 as a CMS2 biomarker in CRC, which is a canonical subtype characterised by WNT and MYC activation. It is important to note that the samples collected and analysed in the CMS classification were mainly from TNM stage II and III tumours compared to samples from healthy tissue. Furthermore, only a small proportion of samples represented metastatic disease [3,37].

Interestingly, several lines of evidence for the potential role of Hakai in cell cycle and stemness have emerged in recent years. First, Hakai affects several cellular phenotypes independent of its effect on E-cadherin. Indeed, the overexpression and knockdown of Hakai affects cell proliferation events in cells that do not express E-cadherin. Hakai overexpression plays an important role in the transition from the G0/G1 to S phase and modulates the protein expression level of the cell cycle regulator Cyclin D1, indicating its role in the cell cycle pathway [20]. Second, the expression level of Hakai is statistically significantly different in adenoma tissues compared to healthy human colon tissues. Adenoma is a benign tumour that arises as the result of proliferative dysplasia, highlighting the role of Hakai at early stages. Third, it is proposed that the post-translational downregulation of E-cadherin is an early event during the EMT, whereas the loss of E-cadherin at the transcriptional level occurs in a late event in the EMT [5,38]. All of these statements are consistent with the implication of Hakai in early stages of tumour growth and carcinoma progression [21]. The canonical CMS2 subtype is characterised by an activation of the WNT and MYC pathways, which are involved in the maintenance and growth of cancer stem cells. Given that EMT has been shown to regulate the acquisition of stem cell-like properties, Hakai may play a role in CSCs. Indeed, a clear association of CMS2 was found for CBLL1 but also for WNT target genes such as LGR5 and C-MYC. In addition, a higher expression of Hakai and Lgr5 proteins was detected in cancer stem cell tumourspheres, while CBLL1 silencing reduces the tumoursphere size and downregulates the Lgr5 biomarker. LGR5 is a WNT target gene that regulates self-renewal capacity and is widely used as a biomarker of CSCs in different types of cancer, including CRC [39]. As previously mentioned, CMS2 is characterised by an activation of Wnt/β-catenin signalling, which is essential in the regulation of CSCs [3,4], and our results support that CBLL1/Hakai plays a role in CSC tumourspheres. In 2018, the first evidence suggesting a potential role for Hakai in stemness was reported [40]. Hakai is involved in m6A methylation, which is the most prevalent and conserved modification at the post-transcriptional regulatory level in eukaryotes. It is well known that Hakai interacts with m6A-binding proteins such as Zc3h13, WTAP and Virilizer [41,42,43]. In 2018, Wen et al. demonstrated that Zc3h13 retains the Zc3h13–WTAP–Virilizer–Hakai complex in the nucleus to regulate m6A methylation. Furthermore, the authors propose that the assembly of this complex in the nucleus is important for conferring self-renewal to mouse embryonic stem cells. Importantly, the depletion of Hakai impairs self-renewal and triggers differentiation. Later, an important study on lung cancer supported the implication of Hakai in stemness through reversing Hakai-mediated EMT. A decrease in E-cadherin and increase in Hakai levels were observed in clinical samples of non-small cell lung cancer with acquired resistance to gefitinib and knockdown of Hakai-elevated E-cadherin expression and attenuated stemness of cancer cells, which resensitised the cells to gefitinib [44]. At present, the molecular mechanism by which Hakai plays a role in colon cancer stem cells is still unknown, and further studies are required to investigate the potential impact on Wnt signalling pathway.

Guinney et al. showed that CMS4 tumours are the most aggressive subtype of all CMSs and that these patients have a worse OS. In contrast, they showed that patients with CMS2 tumours have the best prognosis. However, in-depth analyses of specific biomarkers within a particular CMS subtype could affect clinical decisions. Indeed, in this paper, we analysed the potential of CBLL1 expression to predict OS. When we analysed the CBLL1 mRNA expression considering the whole cohort of CRC patients (including all CMSs), we found that higher CBLL1 expression was associated with longer OS, further supporting the fact that patients with CMS2 tumours have a good prognosis. However, when we grouped CMS2 patients according to high and low CBLL1 expression, high CBLL1 expression is associated with shorter OS and a poor prognosis within CMS2 patients. The same behaviour was also found for the best established biomarker of CSCs in CRC, LGR5. Accordingly, a meta-analysis study showed that higher expression of the CSC biomarker LGR5 was associated with poor prognosis in CRC patients, supporting the need for a better understanding of the biological characteristics of different colorectal tumours required to guide therapeutic decisions [15]. More importantly, CMS2 patients with high CBLL1 expression and poor survival showed the worst prognosis (as observed for CMS4 tumours), highlighting CBLL1 expression as a marker of CMS2 patients with truly poor prognosis. We have previously shown that the Hakai protein gradually increases in TNM stages (I–IV) of colorectal cancer patients compared to adjacent healthy colon tissue [21], while no significant differences were found at the mRNA expression level (Appendix A). This differential association between changes in mRNA and protein levels is well understood by the key role of the post-transcriptional regulation in cancer [45]. Consistent with our results, significant differences between mRNA and protein expression have been found using high-throughput technologies when comparing primary and metastatic samples [46]. In addition, the side of the colon from which the tumour originates also determines OS and prognosis. Indeed, tumours in the right side of the colon have a worse prognosis than tumours in the left side of the colon. CBLL1 is significantly upregulated in left-sided tumours, which are associated with a better prognosis. Indeed, CMS2 tumours were mainly left-sided [3]. Interestingly, although its higher expression is associated with low-risk CRC (Figure 1d), CBLL1 is significantly upregulated in left-sided CRC patients, who tend to have liver and lung metastasis [31,35]. Differences between sides may be due not only to anatomical and developmental origins but also to differences in the microbiota on the two sides of the colon [47]. Therefore, it is important to consider CMS as a classification of samples with different characteristics, which may reflect intratumoural heterogeneity or a transitional phenotype rather than a static biological state.

## 4. Materials and Methods

### 4.1. HT29 and HCT116 Monolayer Cell Culture

The HCT116 human adenocarcinoma colorectal cell line (RRID:CVCL_0291) was cultured in Dulbecco’s Modified Eagle Medium (DMEM) and HT29 human adenocarcinoma colorectal cell line (RRID:CVCL_0320) was cultured in McCoy’s 5A (Modified) Medium (GibcoTM, Waltham, MA, USA). The monolayer cultures were performed in 100 mm dishes (Corning^®^ Costar^®^, New York, NY, USA). Both mediums were supplemented with 10% of foetal bovine serum (FBS, Corning^®^, New York, NY, USA), 100 U/mL penicillin and 100 μg/mL streptomycin at 37 °C under a humidified atmosphere with 5% CO_2_. Both cell lines were purchased from ATCC. Cells were monthly tested for mycoplasma, and no contamination was detected in any cultures. The development of a doxycycline-inducible lentiviral system for shCBLL1 was generated and maintained as previously reported [27]. HT29 sh-Control and HT29 shCBLL1 cells were cultured in McCoy’s 5A (Modified) Medium (GibcoTM, Waltham, MA, USA) supplemented with 10% of foetal bovine serum (FBS, Corning^®^, New York, NY, USA), 100 U/mL penicillin and 100 μg/mL streptomycin. The expression of the lentiviral vector with the small harping was induced with doxycycline (1 µg/mL) (doxycycline hyclate, Sigma Aldrich, St. Louis, MO, USA). Images were taken by using a confocal microscope Eclipse TS100 (Nikon, New York, NY, USA) with a 10× magnification objective.

### 4.2. Cancer Stem Cell Tumoursphere-Derived HCT116 and HT29 Cell Lines

HCT116 and HT29 tumourspheres were cultured using Ultra-Low Attachment 6-well plates (Corning^®^ Costar^®^, NY, USA) in Gibco Dulbecco’s Modified Eagle Medium: Nutrient Mixture F-12 (DMEM/F-12, GibcoTM, Waltham, MA, USA) supplemented with 2% B27 Supplement (Life Technologies, Waltham, MA, USA), 1% GlutaMAX™-100× (GibcoTM, Waltham, MA, USA), 0.04% of 83.3 µM human Epidermal Growth Factor (hEGF, Sigma Aldrich, St. Louis, MO, USA) and 0.1% of 6.25 µM human Fibroblast Growth Factor-Basic (hBFGF, Sigma Aldrich, St. Louis, MO, USA). In both cases, 105 cells were seeded per MW6 well. Images were obtained using a confocal microscope Eclipse TS100 (Nikon, New York, NY, USA) with a 10× magnification objective.

### 4.3. Real-Time Quantitative PCR (RT-qPCR) and Western Blot Analysis

RNA was extracted using TriPure Isolation Reagent (Roche Diagnostics GmbH, Basel, Switzerland) following the instructions provided by the manufacturer. RT-PCR was performed using an NZY First-Strand cDNA Synthesis Kit (MB12502, NZYTech, Lda.—Genes and Enzymes, Lisbon, Portugal). The qPCR was completed using LightCycler^®^ 480 SYBR^®^ Green I Master (Roche Diagnostics GmbH, Basel, Switzerland) and LightCycler^®^ 480 Multiwell Plate 96, white (Roche Diagnostics GmbH, Basel, Switzerland). All the assays were performed in triplicate. The following primer sequences were used: CBLL1 F: 5′CTGGATCCTTGGGTGGTCTT 3′ and R: 5′AGTTCTTTGAGTTCGCGGTG 3′; LGR5 F: 5′AGCAAACCTACGTCTGGAC 3′ and R: 5′ACAGAGGAAAGATGGCAGT 3′; NANOG F: 5′CAGTCTGGACACTGGCTGAA 3′ and R: 5′CTCGCTGATTAGGCTCCAAC 3′; SOX2 F: 5′AACCCCAAGATGCACAACTC 3′ and R: 5′GCTTAGCCTCGTCGATGAAC 3′; KLF4 F: 5′ACTCGCCTTGCTGATTGTCT 3′ and R: 5′GGCCGATCCTTCTTCTTT 3′; c-MYC F: 5′GTTATCTCGCAAACCCCAGA 3′ and R: 5′ACAGAATGGGTCCAGATTGC 3′. As housekeeping controls, we used HPRT F: 5′CCTGGCGTCGATTAGTGA 3′ and R: 5′TCTCGAGCAAGACGTTCAGT 3′ and RPLP0 F: 5′ATGGCAGCATCTACAACCCT 3′ and R: 5′TTGGGTAGCCAATCTGCAGA 3′. Data analysis was performed by using qBasePLUS (version v.3.4) analysis software with GraphPad Prism (version 9.0.2) (Biogazelle, Zwijnaarde, Belgium). Western blot analysis were performed as previously described [27]. The following primary antibodies were used: anti-HAKAI (#A302-969A, 1:1000, Bethyl Laboratories, Montgomery, TX, USA), anti-LGR5 (#PA5-87974, 1:1000, Invitrogen, Carlsbad, CA, USA), anti-Sox-2 (#sc-365823, 1:500, Santa Cruz Biotech., Dallas, TX, USA), anti-NANOG (#sc-374001, 1:1000, Santa Cruz Biotech., Dallas, TX, USA), anti-KLF4 (#MA5-38309, 1:1000, Invitrogen, Carlsbad, CA, USA). As loading control, anti-Vinculin antibody (1:1000, Cell Signaling Technology, Leiden, The Netherlands) was used.

### 4.4. Data Integration and Analysis Using Bioinformatic

We used a dataset consisting of a large cohort of 1273 primary tumour samples from patients with colorectal cancer for which we had full transcriptomic gene expression profiling and clinical information about the overall survival of each patient. This dataset also contains multiple types of phenotypic information from the patients that were considered as covariables in our analysis (such as gender, location of the tumour, and stage, among others). This cohort of CRC samples is an integrated dataset that was produced previously as described in [28]. Using this dataset, we ran two machine learning methods to achieve the reliable classification of each tumour sample into one of the four CRC consensus molecular subtypes (CMS 1, 2, 3 and 4), as defined by Guinney et al. [3]. The classification algorithms applied were CMSclassifier, described in [3], and CMScaller, which is described in [29]. This analysis provided a robust and significant classification of 849 samples of the CRC cohort into a specific CMS (i.e., classification of each CRC tumour sample to a CMS with adjusted *p*-value < 0.05). Therefore, this subset of the cohort was the one used for further comparison of the expression of different genes between CM subtypes.

### 4.5. Statistical Methods

For the experimental data presented in Figure 4 and Figure 5, we used GraphPad Prism 9 software (GraphPad, La Jolla, CA, USA). Statistical significance in the RT-qPCR data was assessed by one-way analysis of variance (ANOVA) and at the indicated significance levels. All experiments were performed in triplicate, and quantitative data are presented as the mean ± standard deviation (SD) or mean ± standard error of mean (SEM).

Using the clinical information from the CRC samples, we performed survival analysis considering the overall survival (OS) time for each sample and the information about the status of each patient (i.e., whether they have just left the study or suffered exitus, which was the positive event considered). The analysis was performed applying Cox logistic regression models, either univariate when analysing one feature (i.e., one gene) or multivariate when analysing multiple features (i.e., multiple genes), which was followed by Kaplan–Meier analysis and graphic representation. Several R packages were used for these analyses: survival, rbsurv, survcomp and uniCox (all available in R-cran: https://cran.r-project.org/, accessed on 1 January 2023). Using these R libraries and packages, we developed an R workflow that incorporated features for survival analysis and associated patient stratification based on the expression level of certain genes tested. In this way, we evaluated the ability of specific genes to mark survival differences considering their expression level in different tumour samples. This method is based on a bootstrapped version of the Kaplan–Meier estimator and the long-rank statistic, which overcomes the instability and irreproducibility of other tools. Boxplots were used to represent the data distributions of the tumour samples considering different molecular or phenotypic features. The calculation of *p*-values in the comparison of the distributions presented in these plots was obtained using a Wilcoxon test.

## 5. Conclusions

In this study, we show that the CBLL1 gene is a biomarker for CMS2 colorectal cancer, as we demonstrate a significant upregulation in this molecular subtype of CRC. We also report for the first time that the Hakai protein (CBLL1 gene) is associated with colorectal cancer stem cells. This pattern extends to the cancer stem cell biomarkers LGR5 and the transcription factor c-MYC, which are universally associated with cancer stem cells (CSCs). Importantly, the high CBLL1 expression in CMS2 patients correlates with shorter overall survival and poor prognosis. Taken together, these findings suggest that CBLL1 is emerging as a potential biomarker for a specific subtype of colorectal cancer tumours and may help to stratify patients based on comparative survival analysis.

## Figures and Tables

**Figure 1 ijms-25-01919-f001:**
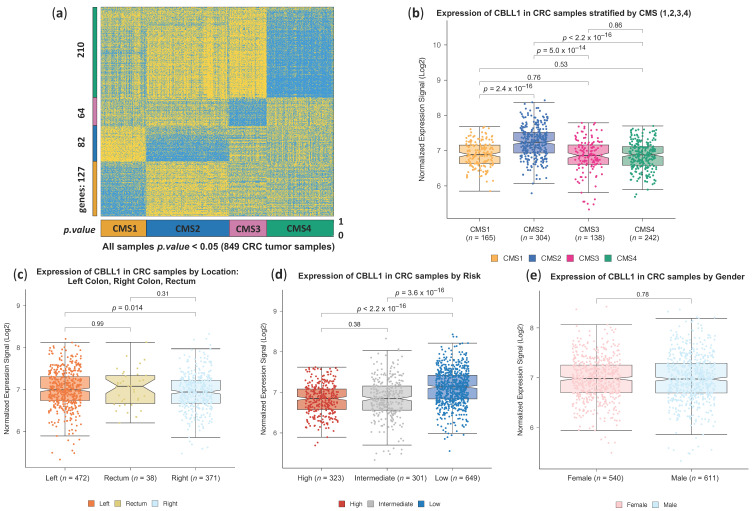
CBLL1 gene expression increases in CMS2 subtype tumour samples of CRC patients. (**a**) Heatmap showing the expression profile of 483 genes in 849 CRC tumour samples classified as significant in one of the 4 consensus molecular subtypes (CMS 1,2,3,4) of colorectal cancer. The classification was performed on an initial cohort 1273 CRC primary tumour samples using the gene signature published by [3], which includes 127 genes for CMS1, 82 genes for CMS2, 64 genes for CMS3, and 210 genes for CMS4; (**b**) boxplots presenting the expression profile of gene CBLL1 (HAKAI) in the four CMSs of 849 CRC tumour samples. The samples are separated into consensus molecular subtypes: CMS1 165 samples, CMS2 304 samples, CMS3 138 samples, and CMS4 242 samples; (**c**) expression of CBLL1 in CRC tumour samples separated by location (for the samples of the 1273 CRC cohort where the location was known): left side of the colon (472 samples), right side of the colon (371 samples), rectum (38 samples); (**d**) expression of CBLL1 in the 1273 CRC cohort separated by risk (low-risk, intermediate and high-risk) following the tumour risk analysis provided in [30]; (**e**) expression of CBLL1 in CRC tumour samples separated by gender: 540 females and 611 males.

**Figure 2 ijms-25-01919-f002:**
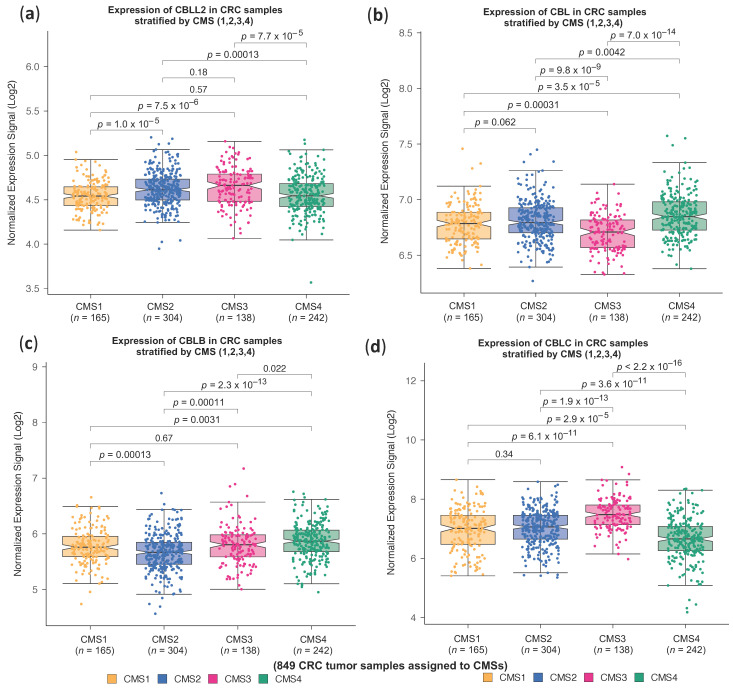
Expression profiles in the cohort of 849 CRC tumour samples of 4 genes that are paralogs of CBLL1. The samples are separated in consensus molecular subtypes: CMS1 includes 165 samples; CMS2 304 samples; CMS3 138 samples; and CMS4 242 samples. Boxplots presenting the expression profile of 4 genes in the four CM subtypes: (**a**) CBLL2; (**b**) CBL; (**c**) CBLB; (**d**) CBLC.

**Figure 3 ijms-25-01919-f003:**
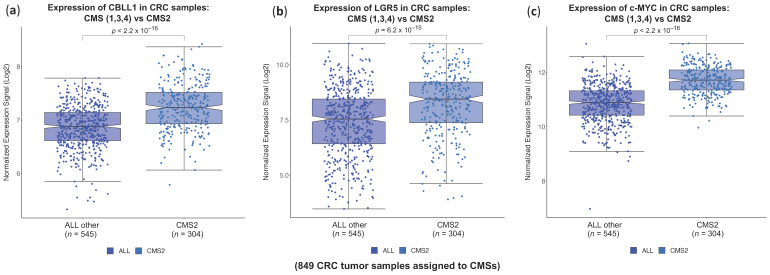
Expression of CBLL1, LGR5 and c-MYC in a cohort of 843 CRC primary tumour samples. The samples are separated in two groups of CM subtypes: CMS2 (*n* = 304 samples) versus the rest of CMSs (CMS1, CMS3, and CMS4, *n* = 545 samples). (**a**) Boxplots presenting the expression of CBLL1 in these two groups; (**b**) boxplots presenting the expression of LGR5 in the two groups; and (**c**) boxplots presenting the expression of c-MYC in the two groups.

**Figure 4 ijms-25-01919-f004:**
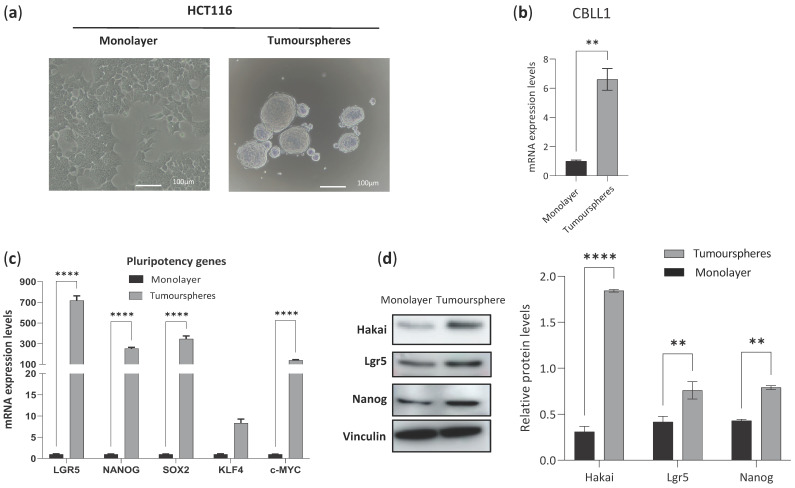
Stem-cell cancer markers expression in HCT116 colon tumour cells in 2D monolayer and 3D tumoursphere. (**a**) Phenotype of HCT116 grown in 2D monolayer and 3D tumoursphere. Optical microscopy images taken after 48 h in culture. Scale bar 100 μm; (**b**) HAKAI mRNA expression and (**c**) mRNA levels expression of 5 genes that mark stem cell (LGR5, NANOG, SOX2, KLF4, c-MYC) in HCT116 grown 2D monolayer and 3D tumoursphere. RPL13A mRNA levels was used as loading control. (**d**) HAKAI and stem cell markers protein levels (LGR5, NANOG) in HCT-116 cells grown in 2D monolayer and 3D tumoursphere. Relative quantification of the protein expression levels is represented in the right panel (** *p* < 0.01; **** *p* < 0.0001). Vinculin was used as protein loading control.

**Figure 5 ijms-25-01919-f005:**
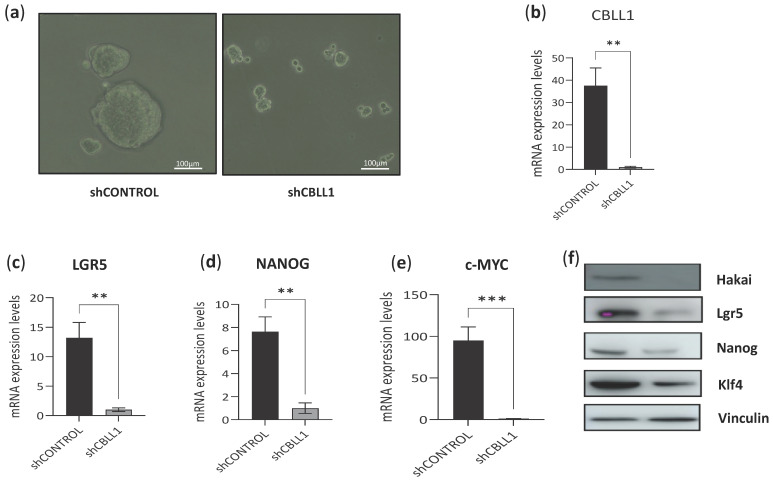
Effect of CBLL1 silencing in HCT116 tumourspheres and cancer stem cell markers. (**a**) Effect of Hakai on phenotype by using shRNA-CBLL1 silencing in an inducible viral system in HT29 cells grown as 3D tumoursphere Optical microscopy images taken after 72 h induction. Scale bar 100 μm. Graphs presenting the mRNA expression levels of: (**b**) HAKAI, (**c**) LGR5, (**d**) NANOG, and (**e**) c-MYC (measured as stem cell markers in an inducible viral system in HT29 cells grown as 3D tumoursphere, ** *p* < 0.01; *** *p* < 0.001); (**f**) HAKAI and stem cell markers protein expression determined by Western blotting in shRNA-CBLL1 silencing in an inducible viral system in HT29.

**Figure 6 ijms-25-01919-f006:**
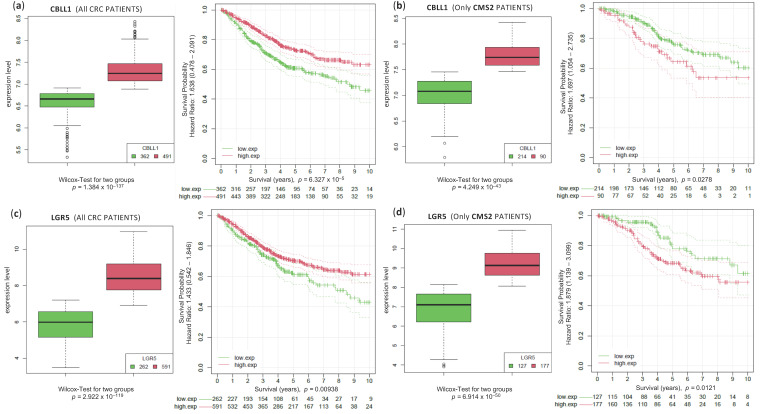
Survival analysis of two genes (CBLL1 and LGR5) in separated CRC cohorts using the expression level to divide each cohort into two populations of high and low expression. (**a**) Boxplots presenting the expression of CBLL1 in CRC tumour samples separated in two groups by expression level (high 491 samples, red, and low 362 samples, green); and the corresponding Kaplan–Meier curve presenting the survival analysis in these two groups of samples. This corresponds to the survival analysis using CBLL1 expression in all 843 patients (i.e., all subtypes CMS1, 2, 3 and 4); (**b**) boxplot presenting the expression of CBLL1 in CRC tumour samples separated by high expression (90 samples, red) and low expression (214 samples, green); and the corresponding Kaplan–Meier curve presenting the survival analysis in these two groups of samples. The second survival analysis is conducted only on the CRC samples of CMS2 subtype. For the other figures: (**c**) is the same as (**a**) and (**d**) is the same as (**b**) but using the expression signal of gene LGR5.

**Figure 7 ijms-25-01919-f007:**
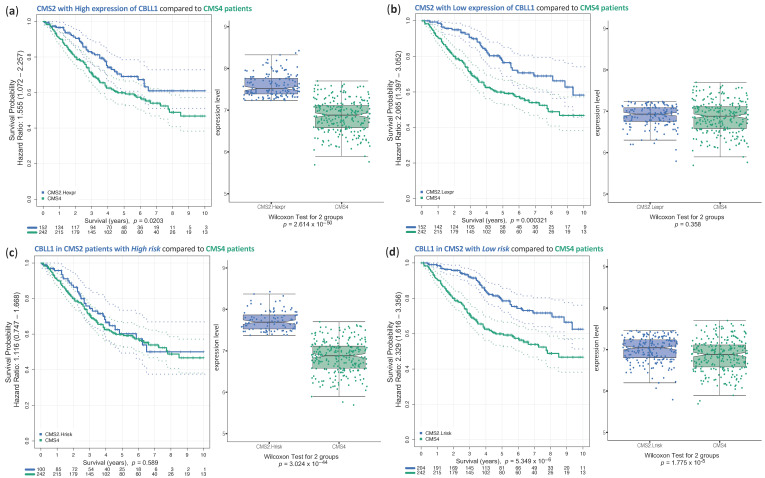
Survival analysis of colorectal CMS2 patients separated using high and low expression (H.expr or L.expr) of the CBLL1 gene (**a**,**b**), as well as separated in patients of high and low risk (H.risk or Low.risk) (**c**,**d**), compared with CMS4 patients. (**a**) Boxplots presenting the comparison of CMS2 patients with high CBLL1 expression (*n* = 152) and CMS4 patients (*n* = 242); and the corresponding Kaplan–Meier curve presenting the survival analysis in these two groups of samples; (**b**) boxplots presenting the comparison of CMS2 patients with low CBLL1 expression (*n* = 152) and CMS4 patients (*n* = 242); and the corresponding Kaplan–Meier curve presenting the survival analysis in these two groups of samples; (**c**) boxplots presenting the comparison of the expression of CBLL1 in CMS2 patients that have high risk (i.e., poor survival) (*n* = 100) and CMS4 patients (*n* = 242); and the corresponding Kaplan–Meier curve presenting the survival analysis in these two groups of samples; (**d**) boxplots presenting the comparison of the expression of CBLL1 in CMS2 patients that have low risk (*n* = 204) and CMS4 patients (*n* = 242); and the corresponding Kaplan–Meier curve presenting the survival analysis in these two groups of samples.

## Data Availability

The data that support the findings of this study are available on request from the corresponding authors (Angélica Figueroa and Javier de las Rivas).

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
