# Peer review of "Stratification of Colorectal Patients Based on Survival Analysis Shows the Value of Consensus Molecular Subtypes and Reveals the CBLL1 Gene as a Biomarker of CMS2 Tumours"

_ijms, 2024, doi:10.3390/ijms25031919_

Round 1
Reviewer 1 Report
Comments and Suggestions for Authors
The current work focuses on the stratification of colorectal patients based on survival analysis shows the value of Consensus Molecular Subtypes and reveals the CBLL1 gene as a biomarker of CMS2 tumors. The author’s great effort into the manuscript, but minor issues should be addressed.
Major issues
-The introduction provides sufficient background and all relevant references are included, but the novelty of this work is not highlighted and the author's contribution was unclear compared to other previous works.
- Conclusion, this section should have more details on the findings.
Minor issues
-Line 40-43 “Given that the molecular characteristics are crucial for predictive prognosis and drug response and that the current classifications give no answers to the clinical needs [2], it became crucial to propose a molecular classification for CRC in order to facilitate the translation of molecular characteristics into the clinic.” Very long sentence try to rephrase it with the same meaning
-Fig. 5a, scale bare should be clear
-Fig. 6. 7 needs improving for easier reading by the reader
Comments on the Quality of English LanguageMinor editing of the English language is required
Author Response
January 17th, 2024
Manuscript ID: ijms-2794203
We appreciate the reviewer for their constructive comments. Please find a revised version of our manuscript Manuscript ID: ID: ijms-2794203 in which we have addressed the reviewer concerns.
POINT-BY-POINT RESPONSE TO THE REVIEWER 1
Thank you to the reviewer for the constructive concerns. As the reviewer suggested, we have clarified the questions or concerns could with minor revisions:
Major issues
-The introduction provides sufficient background and all relevant references are included, but the novelty of this work is not highlighted and the author's contribution was unclear compared to other previous works.
- Thank you for the reviewer suggestion. As the reviewer suggested, in the introduction we have highlighted the novelty of this work and our important contribution in this field (page 3, lines 113-117).
- Conclusion, this section should have more details on the findings.
- Following the reviewer indications, we have revised the conclusion section and we have focused more in our findings (page 15, lines 570-576).
Minor issues
-Line 40-43 “Given that the molecular characteristics are crucial for predictive prognosis and drug response and that the current classifications give no answers to the clinical needs [2], it became crucial to propose a molecular classification for CRC in order to facilitate the translation of molecular characteristics into the clinic.” Very long sentence try to rephrase it with the same meaning
-Thank you for the careful correction of the manuscript. As this reviewer suggested, we have rephrase the indicated sentence in shorter manner (Page 1, lines 39-42).
-Fig. 5a, scale bare should be clear
-Thank you for the careful correction, we have properly included the scale bar in Fig. 5a.
-Fig. 6. 7 needs improving for easier reading by the reader.
As this reviewer suggested, we have improved figures 6 and 7.
(x) Minor editing of English language required
- As you can see in the new version, we have revised the English language.

Reviewer 2 Report
Comments and Suggestions for Authors
The manuscript entitled “Stratification of colorectal patients based on survival analysis shows the value of Consensus Molecular Subtypes and reveals CBLL1 gene as a biomarker of CMS2 tumors” by Alfonsín et al. describes a significant contribution to the stratification system for patients of colorectal cancer (CRC) based on consensus molecular subtypes (CMS). The authors use a combination of bioinformatics analysis of a large series of data from CRC patients and in vitro studies using CRC cell lines and tumourspheres. Their results on the role of CBLL1 is especially interesting, because, as the authors mention, “this is the first reported evidence showing that Hakai protein is associated to colon cancer stem cell”.
The manuscript is well written and the conclusions are soundly based. Nonetheless, some minor issues have been observed in reviewing the submitted text. They are listed below.
1. To describe the results of Fig. 4D, the authors say: “This increase of cancer stem cell biomarkers was confirmed at protein level, as observed in cancer stem cell tumorspheres for Hakai (CBLL1), LGR5, NANOG and SOX2 compared to protein expression in monolayer cultures”. The visual inspection of the figure does not substantiate this statement, especially in the case of CBLL1. Actually, the increase observed for this protein is sensibly similar to that in the loading control. A semi-quantitative evaluation of the band intensity, done with any of the available procedures, may aid to accept or discard the above statement.
2. In the experiments involving CBLL1 knocking down, what is the reason to use spheroids derived from HT29 and not those derived from HCT116? The authors should explain the reason. Better still, although the results shown are clear, comparing the behaviour of stem-cell tumourspheres from both, HCT116 and HT29 would add credibility to these results.
3. The algorithm mentioned on lines 264-5 must be detailed.
Comments on the Quality of English LanguageAlthough the quality of English is good, there are two questions concerning it. First, the text is written using the American spelling, but in some words (e.g., “characterised”, line 136; “personalised”, line 564) the British spelling is used. It would be desirable to use a uniform style. Secondly, there are some typos, which may be corrected. For instance, “closet” instead of closest in line 171, “gropus” for groups in line 265.
Author Response
January 17th, 2024
Manuscript ID: ijms-2794203
We appreciate the reviewer for their constructive comments. Please find a revised version of our manuscript Manuscript ID: ID: ijms-2794203 in which we have addressed the reviewer concerns.
POINT-BY-POINT RESPONSE TO THE REVIEWER 2
Thank you to the reviewer comments. We have included a new version answering the questions or concerns posed by the reviewer:
Minor issues
- To describe the results of Fig. 4D, the authors say: “This increase of cancer stem cell biomarkers was confirmed at protein level, as observed in cancer stem cell tumorspheres for Hakai (CBLL1), LGR5, NANOG and SOX2 compared to protein expression in monolayer cultures”. The visual inspection of the figure does not substantiate this statement, especially in the case of CBLL1. Actually, the increase observed for this protein is sensibly similar to that in the loading control. A semi-quantitative evaluation of the band intensity, done with any of the available procedures, may aid to accept or discard the above statement.
Thank you for the careful revision by the reviewer. As suggested, we have included a quantification of the protein levels for Hakai, Lgr5 and Nanog (right panel of figure 4D and figure legend figure 4). We have detected a statistical significant increase of theses protein marker in tumorosphere stem cell culture versus monolayer culture further accepting our statement.
- In the experiments involving CBLL1 knocking down, what is the reason to use spheroids derived from HT29 and not those derived from HCT116? The authors should explain the reason. Better still, although the results shown are clear, comparing the behaviour of stem-cell tumourspheres from both, HCT116 and HT29 would add credibility to these results.
This is a very good point posed by the reviewer. Unfortunately, we only have HT29 cells in an inducible system by Doxcicline. Therefore, the silencing of Hakai in HCT1116 was performed in transient transfection conditions. In transient transfection, we were unable to maintain Hakai-silencing in these cells lines under these conditions on which we first transfect and after 48 h we split and induce tumorophere formation.
- The algorithm mentioned on lines 264-5 must be detailed.
As the reviewer suggests, we have included a better explanation of the algorithm, including several references where it has been used. We have also rewritten the paragraph containing lines 258-292 to have a better explanation of the survival analysis performed. You can find the new version of this paragraph including the new references:
Bueno-Fortes S, Muenzner JK, Berral-Gonzalez A, Hampel C, Lindner P, Berninger A, Huebner K, Kunze P, Bäuerle T, Erlenbach-Wuensch K, Sánchez-Santos JM, Hartmann A, De Las Rivas J, Schneider-Stock R. A gene signature derived from the loss of CDKN1A (p21) is associated with CMS4 Colorectal Cancer. Cancers 2021; 14(1):136. doi: 10.3390/cancers14010136. PMID: 35008299.
Herrera M, Berral-González A, López-Cade I, Galindo-Pumariño C, Bueno-Fortes S, Martín-Merino M, Carrato A, Ocaña A, De La Pinta C, López-Alfonso A, Peña C, García-Barberán V, De Las Rivas J. Cancer-associated fibroblast-derived gene signatures determine prognosis in colon cancer patients. Mol Cancer. 2021; 20(1):73. doi: 10.1186/s12943-021-01367-x. PMID: 33926453.
Bueno-Fortes S, Berral-Gonzalez A, Sánchez-Santos JM, Martin-Merino M, De Las Rivas J. Identification of a gene expression signature associated with breast cancer survival and risk that improves clinical genomic platforms. Bioinform Adv. 2023; 3(1):vbad037. doi: 10.1093/bioadv/vbad037. PMID: 37096121.
Comments on the Quality of English Language. Although the quality of English is good, there are two questions concerning it.
Thank you for the careful correction. We have revised the English spelling in the all text.
First, the text is written using the American spelling, but in some words (e.g., “characterised”, line 136; “personalised”, line 564) the British spelling is used. It would be desirable to use a uniform style.
We have unified the text using the same spelling.
Secondly, there are some typos, which may be corrected. For instance, “closet” instead of closest in line 171, “gropus” for groups in line 265.
We have corrected in the text the two words indicated by the reviewer.
